# The histone H4K20 methyltransferase SUV4-20H1/KMT5B is required for multiciliated cell differentiation in Xenopus

Alessandro Angerilli[1,*], Janet Tait[1,*] , Julian Berges[1,2], Irina Shcherbakova[1], Daniil Pokrovsky[1] , Tamas Schauer[1] , Pawel Smialowski[3,4], Ohnmar Hsam[1,5], Edith Mentele[1], Dario Nicetto[1,6] , Ralph AW Rupp[1]

**H4 lysine 20 dimethylation (H4K20me2) is the most abundant histone modification in vertebrate chromatin. It arises from sequential methylation of unmodified histone H4 proteins by the mono-methylating enzyme PR-SET7/KMT5A, followed by conversion to the dimethylated state by SUV4-20H (KMT5B/C) enzymes. We have blocked the deposition of this mark by depleting Xenopus embryos of SUV4-20H1/H2 methyltransferases. In the larval epidermis, this results in a severe loss of cilia in multiciliated cells (MCC), a key component of mucociliary epithelia. MCC precursor cells are correctly specified, amplify centrioles, but ultimately fail in ciliogenesis because of the perturbation of cytoplasmic processes. Genome-wide transcriptome profiling reveals that SUV4-20H1/H2-depleted ectodermal explants preferentially down-regulate the expression of several hundred ciliogenic genes. Further analysis demonstrated that knockdown of SUV4-20H1 alone is sufficient to generate the MCC phenotype and that its catalytic activity is needed for axoneme formation. Overexpression of the H4K20me1-specific histone demethylase PHF8/KDM7B also rescues the ciliogenic defect in a significant manner. Taken together, this indicates that the conversion of H4K20me1 to H4K20me2 by SUV4-20H1 is critical for the formation of cilia tufts.**

## Introduction

Methylation of histone residues is a major means by which the epigenetic regulation of gene expression is achieved. For instance, in higher eukaryotes, trimethylation of histones at H3K9, H3K27, and H4K20 represent the major repressive epigenetic marks, whereas methylated H3K4 and H3K36 are hallmarks of actively transcribed genes (Saksouk et al, 2014; Hyun et al, 2017). In proliferating cells, the maintenance of the epigenetic information is potentially challenged in the S phase, when parental histones are partitioned to the newly duplicated DNA strands, and the local density of their posttranslational modifications (PTMs) being effectively diluted through incorporation of newly synthesized, unmodified histones (Scharf et al, 2009). Recovery of the original level of modifications is usually achieved within one cell cycle, with notable exceptions (Alabert et al, 2015; Petryk et al, 2018). Mono and di-methylation of H4K20 represent a unique case, because these marks are written in separate cell cycle phases (Pesavento et al, 2008; Beck et al, 2012b). Newly incorporated H4 histones become monomethylated on K20 in a genome-wide fashion by PR-SET7/KMT5A, a histone methyltransferase, whose activity is mainly restricted to the G2 and M phases of the cell cycle because of proteolytic degradation in late G1 (Beck et al, 2012a). Thus, H4K20me1 levels reach a maximum in late mitosis before the next G1 phase SUV4-20H1 and SUV4-20H2 enzymes convert this modification to higher methylated states (Schotta et al, 2004). Consequently, H4K20me1 quickly disappears from G1 phase chromatin with exception of some loci, where it is shielded by unknown factors (Jorgensen et al, 2013). The lowest H4K20 monomethyl levels are found in quiescent and differentiated cells which exhibit a shift towards the di and trimethylated states (Biron et al, 2004; Tsang et al, 2011; Evertts et al, 2013). Remarkably, in both Xenopus tadpoles and human cells, H4K20me2 covers ~80% of total histone H4 and thus represents the most abundant histone modification in vertebrates (Evertts et al, 2013; Schuh et al, 2020; Pokrovsky et al, 2021).

H4K20 methylation states have distinct functional connotations. In mammals, H4K20me1 has demonstrated roles in the firing of replication origins (Shoaib et al, 2018), maintenance of genome integrity (Oda et al, 2009), and chromosome condensation (Beck et al, 2012a). Its role in transcriptional regulation appears to be

[1]Department of Molecular Biology, Biomedical Center, Ludwig-Maximilians-Universität München, Planegg-Martinsried, Germany   [2]Sektion Pädiatrische Pneumologie und Allergologie und Mukoviszidose-Zentrum, Universitäts-Klinikum Heidelberg, Heidelberg, Germany   [3]Institute for Stem Cell Research, Helmholtz Centre Munich, Neuherberg, Germany   [4]Department of Physiological Genomics, Biomedical Center, Ludwig-Maximilians-Universität München, Planegg-Martinsried, Germany   [5]Klinik und Poliklinik für Neurologie der Universität Regensburg, Regensburg, Germany   [6]Ambys Medicines, South San Francisco, CA, USA

Correspondence: ralph.rupp@bmc.med.lmu.de
*Alessandro Angerilli and Janet Tait shared first authors

context dependent. H4K20me1 contributes to the down-regulation of X-linked genes during dosage compensation in *C. elegans* (Brejc et al, 2017) and down-regulates genes associated with cytoskeleton organization and cell adhesion in mammals (Asensio-Juan et al, 2012). Other reports link it to gene activation during canonical WNT- and IFNγ-signaling (Li et al, 2011; Shoshani et al, 2014; Asensio-Juan et al, 2017; Zhou et al, 2017). The premature conversion of H4K20me1 into H4K20me2 during the G2/M phase of the cell cycle leads to strong mitotic defects. Functions for H4K20me2 have remained elusive because of its ubiquitous presence (Jorgensen et al, 2013). H4K20me3 is a transcriptional repressor in Xenopus embryos and mouse ES cells (Nicetto et al, 2013; van Kruijsbergen et al, 2017; Kurup et al, 2020). It primarily localizes to centromeres, telomeres, and repetitive DNA elements, and it participates in heterochromatin formation by recruiting factors such as cohesins (Hahn et al, 2013).

Here, we report that depletion of either both SUV4-20H enzymes or SUV4-20H1 alone strongly affects the formation of ciliary axonemes, the major specialized cytoskeleton structure of multiciliated cells (MCCs) on the Xenopus larval epidermis. Taking advantage of epidermal organoid cultures (animal cap explants) we demonstrate that knocking down SUV4-20H enzymes negatively affects the expression of hundreds of cilium and cytoskeleton genes with striking specificity. This study shows that such repression is a consequence of the high abundance of H4K20me1 on postmitotic chromatin. In WT cells, this repressive effect is neutralized by conversion of H4K20me1 to H4K20me2 status through SUV4-20H1. Ultimately, our data establish that SUV4-20H1 enzyme activity is needed for the formation of the multiciliated tuft.

## Results

### SUV4-20H enzymes are required for differentiation of MCCs in Xenopus larval epidermis

In order to study the function of H4K20 methylation in vivo, we induced protein knockdown (KD) of both SUV4-20H enzymes via radial injection of translation blocking antisense morpholino oligonucleotides (Mo) with established specificity and effectiveness (Nicetto et al, 2013) and investigated the consequences on H4K20 methyl marks by quantitative mass spectrometry (Fig 1A). SUV4-20H2 KD had a much stronger effect on H4K20me3 levels than SUV4-20H1 KD (Fig 1A'), similar to what is observed in a mouse. KD of either enzyme also strongly reduced the H4K20me2 levels, with the double KD producing the strongest effect. This indicates that the function of the two enzymes is partially redundant. Interestingly, upon individual or double KD of the SUV4-20H enzymes, we observed a strong enrichment for H4K20me1, indicating that like in mammals, this histone mark serves as a substrate for SUV4-20H methyltransferases in Xenopus.

We noticed that double-morphant (dMO) tadpoles failed to exhibit the typical forward-sliding movement over ground, which is caused by the synchronous and directional beating of motile cilia on the larval epidermis. By ejecting an aqueous dye close to the surface of manipulated embryos, we found that the liquid flow was strongly impaired in SUV4-20H dMO embryos compared with control morpholino (CoMo)-injected embryos (Video 1 and Video 2). A reduced flow reflects either a problem with polarization of the cilia stroke direction (Mitchell et al, 2009) or a dysfunction of the cilia tufts on the skin surface. To distinguish between these possibilities, we injected embryos unilaterally with control or *suv4-20h1/h2* morpholinos and performed whole mount immunostainings against acetylated alpha-tubulin, a major component of ciliary axonemes (Fig 1B and C). The vast majority of SUV4-20H dMOs displayed clearly reduced ciliary staining on their injected side. This phenotype was reproduced with a second pair of *suv4-20h1/h2*-specific Mos (supplementary material, Fig S1B and C), whose target regions are non-overlapping with the first Mo pair (Fig S1A). In addition, cilia tufts were significantly restored by co-injection of morpholino-insensitive Xenopus *suv4-20h1/h2* mRNAs (*P*adj = 0.012), but not by co-injection of lacZ mRNA (*P*adj = 0.776). We also noticed that overexpression of xSUV4-20H1/H2 proteins alone increased the density of MCCs within the epidermis, while lacZ mRNA had no effect (Fig S1D and E).

To better elucidate the molecular features of this phenotype we performed a cell mosaic analysis by injecting a single ventro-animal blastomere at the eight cell-stage, whose descendants become mostly epidermis, and analysed the consequences by confocal microscopy (Fig 1D–G). The injected cells were lineage-traced by hyls1-gfp mRNA, which encodes a widely conserved protein stably incorporated into the outer centriolar wall (Dammermann et al, 2009). In this way, the progeny of the injected blastomere intermingles with the surrounding wt cells and the manipulated MCCs could be identified by GFP-positive basal bodies (BB). We then harvested the embryos at tailbud stage (NF28), that is, after formation of the mucociliary epithelium, and stained them for acetylated alpha-tubulin (cilia), filamentous actin (cell borders/apical actin lattice), and DNA (nucleus). This experiment demonstrated that SUV4-20H1/H2 dMO MCCs are present on the surface of the embryo and produce in deep cytoplasm a large number of centrioles (Fig 1D and E). This is very similar to WT MCCs, which are known to generate hundreds of centrioles at the outer nuclear membrane via the deuterosome pathway (Meunier & Azimzadeh, 2016). A more detailed comparison with CoMo MCCs, however, revealed that the BBs in SUV4-20H1/2-depleted MCCs tend to clump in deep cytoplasm and were delayed in transport to and docking at the apical cell membrane (Fig 1E, compare optical sections #s 2–5 between CoMo and dMOs). Notably, most BBs, which had arrived at the cell membrane, still failed to nucleate ciliary axonemes, detailing a nearly complete loss of cilia in many dMO MCCs. In addition, the apical actin lattice, which forms around BBs (Sedzinski et al, 2016), appears much less dense and uniform in SUV4-20H1/H2 dMO MCCs, compared with CoMo MCCs. The differences in cilia and apical actin staining were highly significant between control and SUV4-20H-depleted MCCs (Fig 1F and G). Together, these observations indicate that SUV4-20H1/H2-depleted MCCs display a differentiation defect that is because of multiple, possibly linked defects in cytoskeleton structures and processes, which are prerequisites for ciliogenesis. The severe reduction in cilia numbers explains the reduced liquid flow observed along the embryonic epidermis.

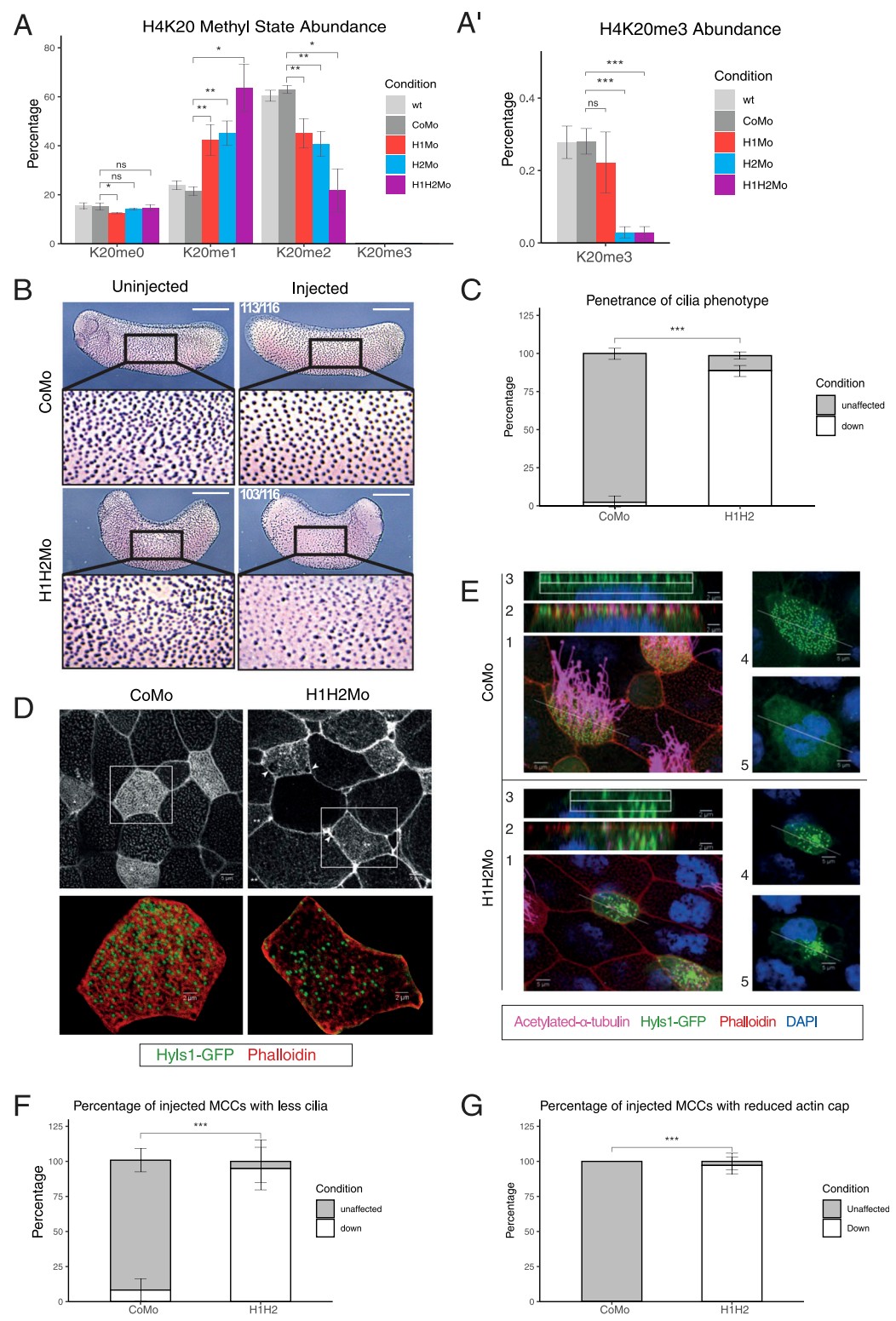

**Figure 1. SUV4-20H1/H2 enzymes are required for ciliogenesis.**
**(A, A')** Relative abundance of H4K20 methyl states by mass spectrometry in *X. laevis* bulk embryonic chromatin upon control (CoMo), single or double KD of SUV4-20H1 and SUV4-20H2 enzymes. **(B, C)** Representative immunocytochemistry images of multiciliated cells (acetylated a-tubulin) in a tail bud stage embryo upon control or SUV4-20H dKD (inserts show enlarged sections of the same images, scale bars = 1 mm) and (C) quantification of (B) (n = 6 biological replicates). **(D, E)** Representative immunofluorescence images detailing the multiciliated cell phenotype upon *suv4-20h1/2* KD. Basal bodies are green (hyls1-GFP), the cilia are magenta (acetylated a-tubulin antibody), the apical actin meshwork is red (phalloidin) and the nucleus is blue (DAPI). **(E)** Confocal analysis of the actin meshwork and docked basal bodies in CoMo or H1H2Mo-injected MCC. Panels show: 1—a representative MCC, 2—overlap between the actin cap and basal bodies on the apical surface, 3—basal bodies in only the

### SUV4-20H enzymes act downstream of MCI and FOXJ1

The multiciliogenic differentiation program begins when an epidermal stem cell gets specified by Notch/Delta signaling as a MCC precursor (Brooks & Wallingford, 2014). Notch regulates in MCC precursors the expression of MULTICILIN/MCIDAS (MCI), a transcriptional coactivator protein, which establishes a core MCC transcriptome of about 800 genes with help from key downstream transcription factors, such as FOXJ1 and RFX2 (Quigley & Kintner, 2017). We therefore wondered whether the effect of SUV4-20H depletion on ciliogenesis could be overcome by *mci* overexpression. We addressed this question in mosaic embryos, co-injecting morpholinos, and *hyls1-gfp* transcripts with synthetic mRNA of a hormone-inducible *mci–hGR* fusion construct (Quigley & Kintner, 2017). We administered dexamethasone at the late gastrula stage (NF11, that is, when the endogenous mci gene gets activated) and harvested the embryos at the late tail bud stage (NF28). In controls, *mci–hGR* induced both de novo amplification of centriole numbers and formation of cilia even in goblet cells, unambiguous evidence for an MCI gain of function phenotype. In SUV4-20H1/H2 dMO embryos, however, MCI-hGR did not restore the apical actin cap or the ciliary axonemes (Fig S2A–C).

To exclude the possibility that MCI could be limited in its function to activate *foxj1*, a transcription factor required to produce motile cilia nucleated from the mother centriole of cells (Walentek et al, 2015), we also overexpressed this transcription factor in the dMO condition. The enhanced levels of FOXJ1 were sufficient to induce ectopic axoneme formation in CoMo goblet cells, but could not increase cilia numbers in SUV4-20H1/H2-depleted MCCs (Fig S2D–F). Taken together, these results indicate that SUV4-20H1/H2 enzymes are essential for multiciliogenesis.

### SUV4-20H1/H2-dependent transcriptional profile of Xenopus epidermis

To analyse the changes in gene expression associated with the SUV4-20H dMO epidermal cells, we decided to take advantage of the animal cap (AC) organoid system. ACs are prospective ectodermal explants that recapitulate the differentiation of the embryonic epidermis. We injected embryos radially at the two-cell stage with either control or *suv4-20h1/h2* Mos. We then dissected ACs at the blastula stage and performed RNA-seq analysis at three key developmental stages (gastrula, neurula, and tailbud) in three biological replicates, obtaining approximately 100 million reads per stage and condition. Previous work from our laboratory has detailed major changes in transcription of differentiating AC cells to occur between gastrula (NF10.5) and neurula (NF16) stages (Angerilli et al, 2018). Consistent with these findings, differential gene expression analysis for the gastrula stage revealed very few genes to be misregulated between CoMo and SUV4-20H dMO conditions (Table S1). At the tail bud stage, we observed that dMO ACs dissociated spontaneously into single cells, indicating a problem with cell adhesion. RNA profiles of these replicates clustered heterogeneously

in principal component analysis, which made it difficult to assess the differentially expressed genes (Table S2). This inclined us to focus further analysis on neurula-equivalent ACs. At this timepoint, MCCs have become postmitotic and differentiate, as manifested by de novo production of centrioles/basal bodies and intercalation into the outer cell layer of the epidermis (Deblandre et al, 1999; Marcet et al, 2011).

We found that the expression of 3,686 genes (19.7% of all genes) were altered in this dataset, most of them (2,246) being downregulated (Fig 2A and Table S3). We then performed gene ontology enrichment analysis for both transcriptional responses to the altered H4K20 methyl landscape. The group of genes that was up-regulated in the SUV4-20H dMO condition was enriched for biological processes involved in chromatin organization and remodeling, methylation, gene expression, and various metabolic processes (Fig 2B). In confirmation of previous results, we found *oct25/pou5f3.2* and *oct91/pou5f3.1* among these genes, which we have shown to be repressed by H4K20me3 deposition (Nicetto et al, 2013). In contrast, for the cohort of genes down-regulated in SUV4-20H dMO ACs, the top 20 entries were all associated with terms connected to cilium, centrosome, microtubules, and cytoskeleton (Fig 2C). The most enriched categories relate to "cytoskeleton" and "cilium" and comprise of hundreds of genes. These are typically expressed at medium to high levels in control ACs, and many of them are down-regulated in SUV4-20H1/H2-depleted chromatin. The significantly down-regulated genes included 369/1603 cytoskeleton genes and 183/440 cilium genes (Fig 2D and E and Table S3). Notably, these two GO terms are biologically related and overlap significantly for all genes in these categories, but also with regard to genes becoming down-regulated in SUV4-20H morphants (Fig S3A and B). Similarly, a strong correlation is found when we compare our results with the "MCC core gene list," which compiles experimentally validated ciliogenic genes downstream of MCI (Quigley & Kintner, 2017). Forty percent of all cilium genes and 70% of the cilium genes down-regulated in *suv4-20h* morphants belong to this list (Fig S3C and D). Therefore, the transcriptome analysis of SUV4-20H-depleted ACs implies mainly a concerted down-regulation of genes that are needed for cilia tuft formation.

Most interestingly, key transcriptional regulators of ciliogenesis were expressed either at normal (*foxj1*, *rfx2*) or slightly up-regulated levels (*mci*) in the dMO condition (Table S3). Thus, SUV4-20H1/H2 enzymes are essential chromatin regulators which allow these transcription factors to execute the program of multiciliogenesis. The widespread down-regulation of genes associated with cytoskeleton structures and cilia provides a robust explanation for the specific MCC phenotype arising from SUV4-20H1/H2 depletion, but how does the altered H4K20 methyl landscape cause such a massive deregulation of ciliogenic gene transcription?

### The histone demethylase PHF8/KDM7B improves ciliogenesis in SUV4-20H dMO epidermis

The H4K20me1 modification is important for mouse development (Schotta et al, 2008; Oda et al, 2009) and is found on promoters,

---

uppermost Z-sections, 4—an apical view of the basal bodies, 5—a deep Z-section close to the cell nucleus. **(F, G)** Quantification of the number of MCCs showing reduced cilia and filamentous actin staining after confocal analysis. We measured 144 CoMo and 163 H1H2Mo-injected multiciliated cells. Error bars represent standard deviations.

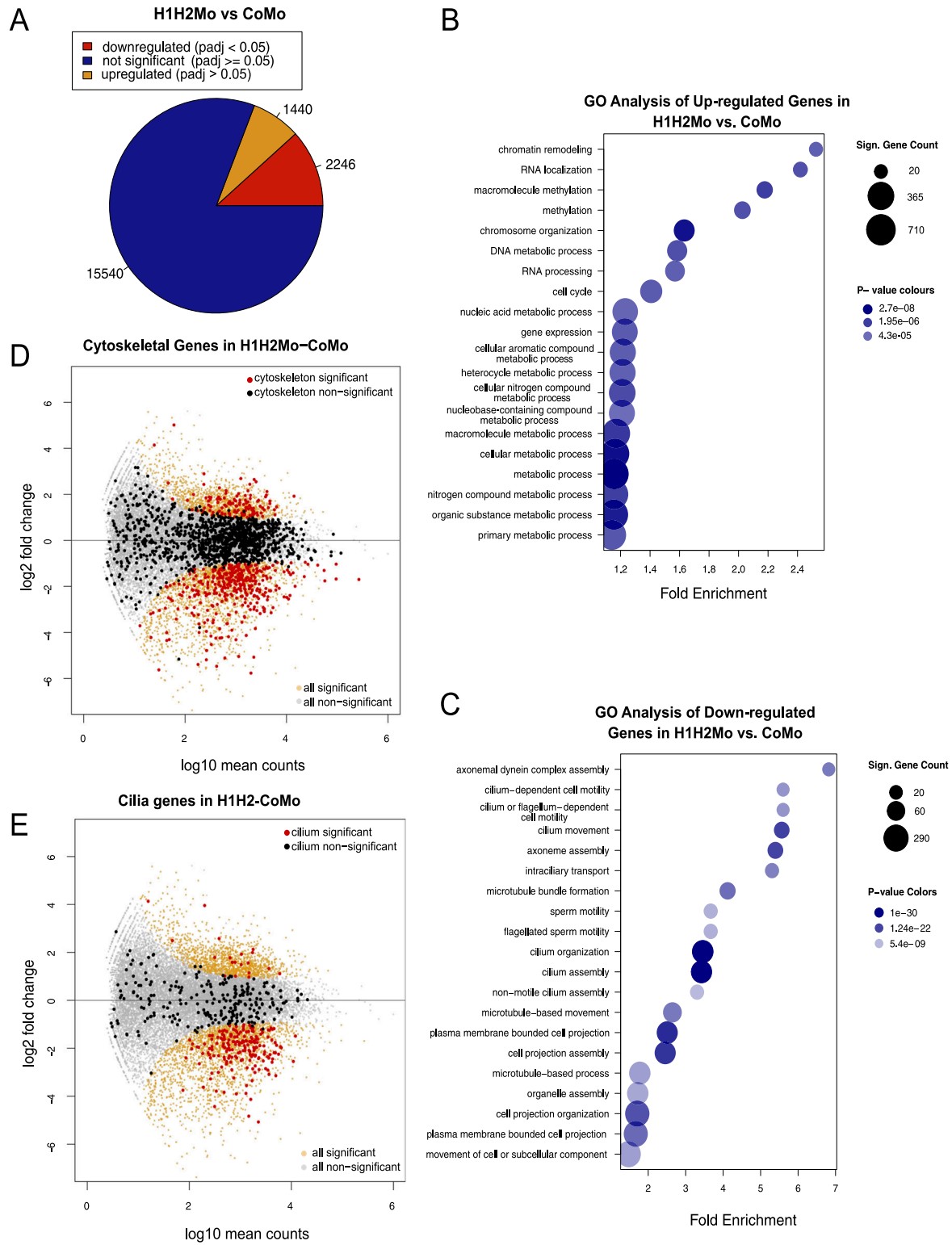

**Figure 2.   The transcriptome of SUV4-20H1/H2-depleted ACs reveals a link to cilogenesis.**
**(A)** Number of misregulated genes in SUV4-20H1/2-deficient ACs (Table S3). **(B, C)** Gene ontology (GO) analysis for up-regulated genes (B) and down-regulated genes (C) in *suv4-20h1/2* morpholino-injected ACs normalized to control morpholino-injected ACs. Bubble size represents the number of significant genes per GO term and bubble colour represents *P*-value. **(C)** GO analysis for down-regulated genes in *suv4-20h1/2* morpholino-injected ACs. Bubble size represents the number of significant genes per GO term and bubble colour represents *P*-value. **(D, E)** MA plots showing gene expression in *suv4-20h1/2* morpholino-injected ACs. Each dot represents a single gene. Nonsignificant genes are indicated in light grey and significant in orange (*P*adj < 0.05). Cytoskeleton genes and cilium genes (as defined by R/Bioconductor package:

coding regions of genes, and in intergenic regions (Barski et al, 2007; Beck et al, 2012b). Whether this mark represses or activates gene transcription has remained under dispute (for a discussion see Beck et al, 2012a; Beck et al, 2012b; Brejc et al, 2017; Huang et al, 2021). As an orthogonal approach to investigate the contribution of H4K20me1 on cilium tuft formation, our attention was attracted to PHF8/KDM7B, a JmjC-domain containing histone demethylase, which is targeted to promoters by interaction of its N-terminal PHD domain with H3K4me2/3. This enzyme has multiple substrates including H3K9me1/2, H3K27me2, and H4K20me1. Several reports demonstrate that knockdown of PHF8 predominantly increases H4K20me1 levels at coding regions and reduces gene expression (Liu et al, 2010; Qi et al, 2010; Asensio-Juan et al, 2017), impacting cytoskeleton organization, cell adhesion, and neurite outgrowth (Asensio-Juan et al, 2012). Based on some similarity between these and our observations (i.e., reduced cell adhesion and cytoskeleton gene expression), we decided to test whether overexpression of PHF8 could improve cilium tuft formation in SUV4-20H dMOs.

We used two *phf8* cDNA clones available from commercial gene repositories: a full-length human clone (hPHF8) and a partial *Xenopus phf8* clone (xPHF8ΔC). The latter consists of the first 264 aa residues of the xphf8 ORF, including the PHD domain known to target the protein to H3K4me3 marks at active promoters, and the first 66 amino acids from the 156-residue long JmjC-domain (Fig S4B). Of the critical amino acid triad His-Asp-His in the catalytic center, the truncated Xenopus protein lacks the third residue involved in Fe-coordination and, thus, is potentially compromised in its catalytic activity (Chaturvedi et al, 2019).

First, we injected hPHF8 mRNA, either alone or in combination with *suv4-20h* Mos, and evaluated its effect on ciliogenesis by staining half-injected embryos for acetylated-$\alpha$-tubulin at the tail bud stage. Injection of hPHF8 alone had little to no effect on the density or size of cilia tufts; however, it restored cilia staining significantly in SUV4-20h dMOs. In contrast, co-injection of the same amount of lacZ mRNA had no effect on the phenotype (Fig 3A and B). In comparable settings, xPHF8ΔC mRNA caused with high penetrance (96%) an increase in MCC numbers and ciliary staining intensity when injected alone (Fig S4A and C). In addition, it restored ciliogenesis in a significant manner, reducing the frequency of embryos with strongly reduced cilia from 90% to 25%. In addition, we confirmed by confocal microscopy in mosaic embryos that both *phf8* mRNAs were able to significantly restore the assembly of axonemes and improve the density of the apical actin meshwork (hPHF8: 85% of MCCs, xPHF8ΔC: 77% of MCCs), although not to the level of neighboring WT MCCs (Figs 3C–E and S4D and E).

These findings stimulated us to investigate the PHF8-dependent changes in gene expression, which accompany this partial morphological rescue. We injected embryos with hPhf8 mRNA alone or in concert with the *suv4-20h1/h2* Mos ("hPHF8-rescue" condition). We also injected CoMo and LacZ mRNA as non-specific controls. The results from hPHF8-, CoMo-, and LacZ-injected explants were highly similar (Fig S5 and Tables S4 and S5). In order to stringently isolate

the rescuing effect of hPHF8 on the *suv4-20h* dMO condition, we normalized the transcriptome of hPHF8-rescue ACs by the transcriptome of *phf8*-overexpressing ACs (Fig 4). In the hPHF8 rescue condition, 2,384 genes were significantly up-regulated, whereas 2,167 genes were significantly down-regulated (Fig 4A and Table S6). Of genes that were down-regulated in the initial *suv4-20h1/2* dMO condition, 1,663 improved their expression and 389 were further down-regulated (Fig 4B). To correlate these changes in gene expression with the observed morphological rescue, we specifically investigated whether transcript levels of cilium and cytoskeleton genes, which were down in the initial SUV4-20H1/2 KD dataset, had improved (Fig 4D and F). Indeed, out of 183 cilium genes, 140 improved their log$_2$-fold change (77%), whereas 43 were further down-regulated; and out of 363 cytoskeleton genes, 290 were improved in their expression (80%), whereas 73 were further down-regulated (Fig 4C and E).

The differential gene expression analysis for the xPHF8ΔC-rescue suggested that this truncated enzyme had a stronger activity on its own than the full-length human protein (Fig S6 and Table S7). By comparison of the log$_2$-fold changes, xPHF8ΔC clearly improved the expression levels of cilium and cytoskeleton genes, which were down in SUV4-20H dMO condition (Fig S6B and D). This was found for 176/183 (96%) cilium genes and 344/363 (95%) cytoskeleton genes (Fig S6A and C).

All together, these results indicate that overexpression of the histone demethylase PHF8 has an influence on gene transcription in SUV4-20H-depleted epidermal organoids. The improved expression levels of cilium and cytoskeleton genes are compatible with the observed partial rescue of cilia tufts on the morphological level. This supports the hypothesis that the ciliogenic defect is caused through gene repression by H4K20me1.

## Cilia tuft formation requires catalytic activity of SUV4-20H1, independently from SUV4-20H2

These last results strongly suggest that it is the increase in H4K20me1 abundance, rather than the loss of the H4K20me3 state, which blocks cilia tuft formation. Although both SUV4-20H enzymes contribute to H4K20me2 levels, the conversion to the trimethyl state (a predominant property of SUV4-20H2; see Fig 1) seems irrelevant. This raised the question, whether indeed both enzymes are required for multiciliogenesis.

To address this, we unilaterally injected embryos with either *suv4-20h1* Mo, *suv4-20h2* Mo or both and determined the effects by immunostaining for cilia tufts. As shown in Fig 5A and B, the KD of SUV4-20H1 alone elicited the loss of cilia to a similar extent and penetrance as the double KD of both enzymes. In contrast, SUV4-20H2 depleted embryos displayed WT-like cilia tufts. We then asked whether cilia tufts require the enzymatic activity of SUV4-20H1. Epidermal organoids, depleted only for SUV4-20H1, were co-injected with mRNAs, encoding either wt or a catalytically inactive Xenopus SUV4-20H1 variant. WT SUV4-20H1 protein restored

org.Mm.eg.db version 3.8.2, mouse annotation) are indicated in black (nonsignificant, *P*adj > 0.05) and red (significant, *P*adj < 0.05). Data normalized to control morpholino-injected animal caps.

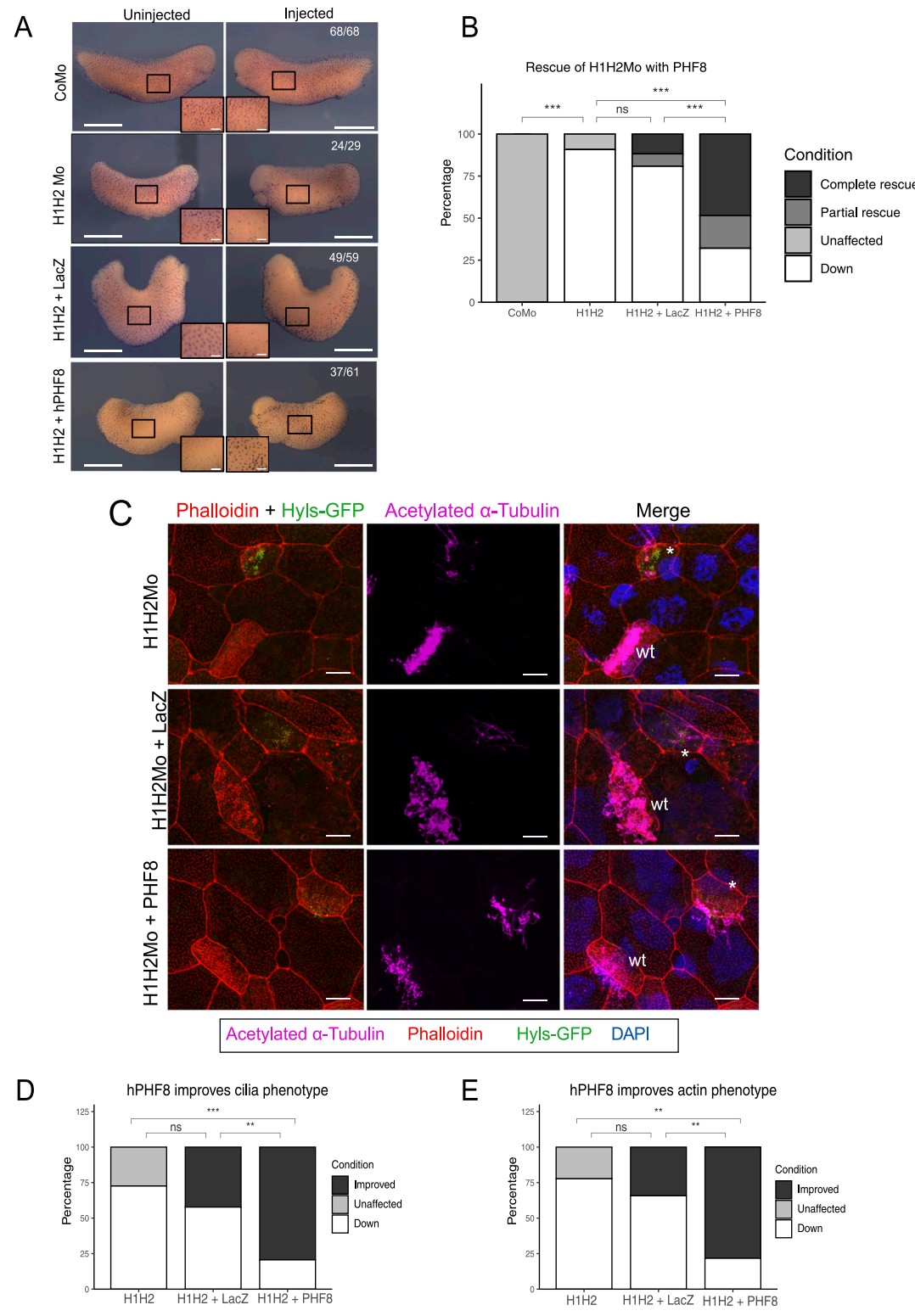

**Figure 3.   Rescue of the ciliogenic phenotype with hPHF8.**
**(A)** Representative immunocytochemistry images of tail bud stage embryos stained for acetylated a-tubulin (multiciliated cells). Injected reagents shown on y-axis, uninjected, and injected sides shown on top. Scale bars = 1 mm (whole embryo) and 200 $\mu$m (inserts), n = 4 biological replicates. **(B)** Quantification of (A).
**(C)** Representative confocal images detailing the multiciliated cell phenotype upon *suv4-20h1/2* KD and both rescue conditions. The basal bodies are green (hyls1-GFP), the ciliary axonemes are magenta (acetylated a-tubulin antibody), the apical actin meshwork is red (phalloidin), and the nucleus is blue (DAPI). A mosaic injection scheme is used allowing KD and wt cells to be present in the same field of view (* = KD MCCs, wt = wildtype MCCs). **(C, D, E)** Quantification of (C). Scale bars = 10 $\mu$m, n = 3 biological replicates.

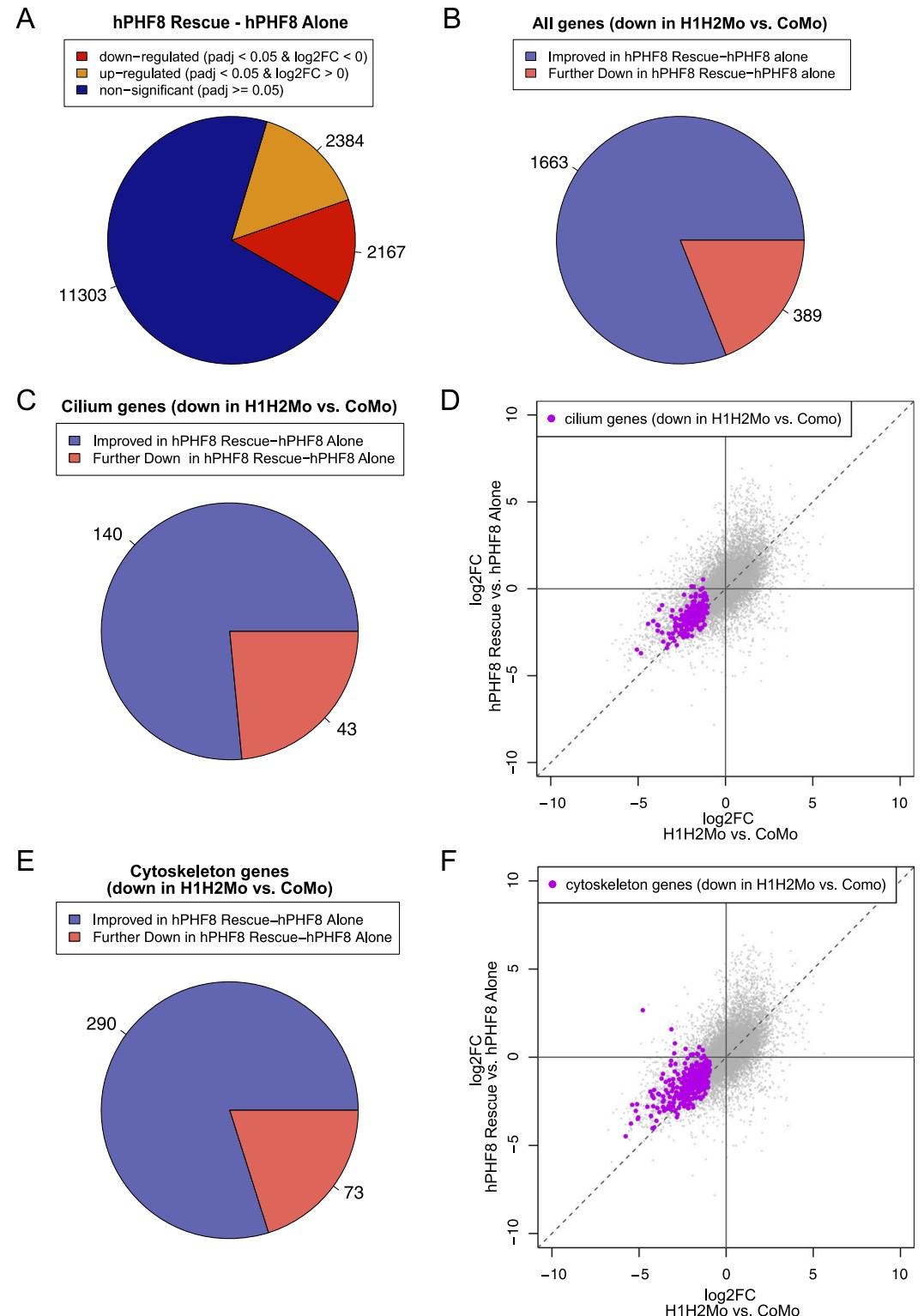

**Figure 4. hPHF8 improves gene expression in suv4-20h1/2 dMO ACs.**
**(A)** Number of misregulated genes in hPHF8 Rescue (H1H2Mo + hPHF8 mRNA injected ACs) normalized to hPHF8-alone ACs. **(B, C, E)** Change in expression of all genes (B), cilium genes (C) or cytoskeleton genes (E) that were down-regulated by SUV4-20H1/2 knockdown upon hPHF8 rescue. To investigate whether transcript levels of cilium and cytoskeleton genes, which were down in the initial SUV4-20H1/2 KD dataset, had improved, we compare the log₂fold change between H1H2Mo versus CoMo-injected animal caps to log₂ fold change of hPHF8 Rescue versus hPHF8-injected animal caps. **(D, F)** shows the result for cilium genes and (F) for cytoskeleton genes. Dashed line indicates no difference in expression upon hPHF8 rescue. Genes above the line have improved expression and genes below the line are further down-regulated. (n = 3 biol. replicates).

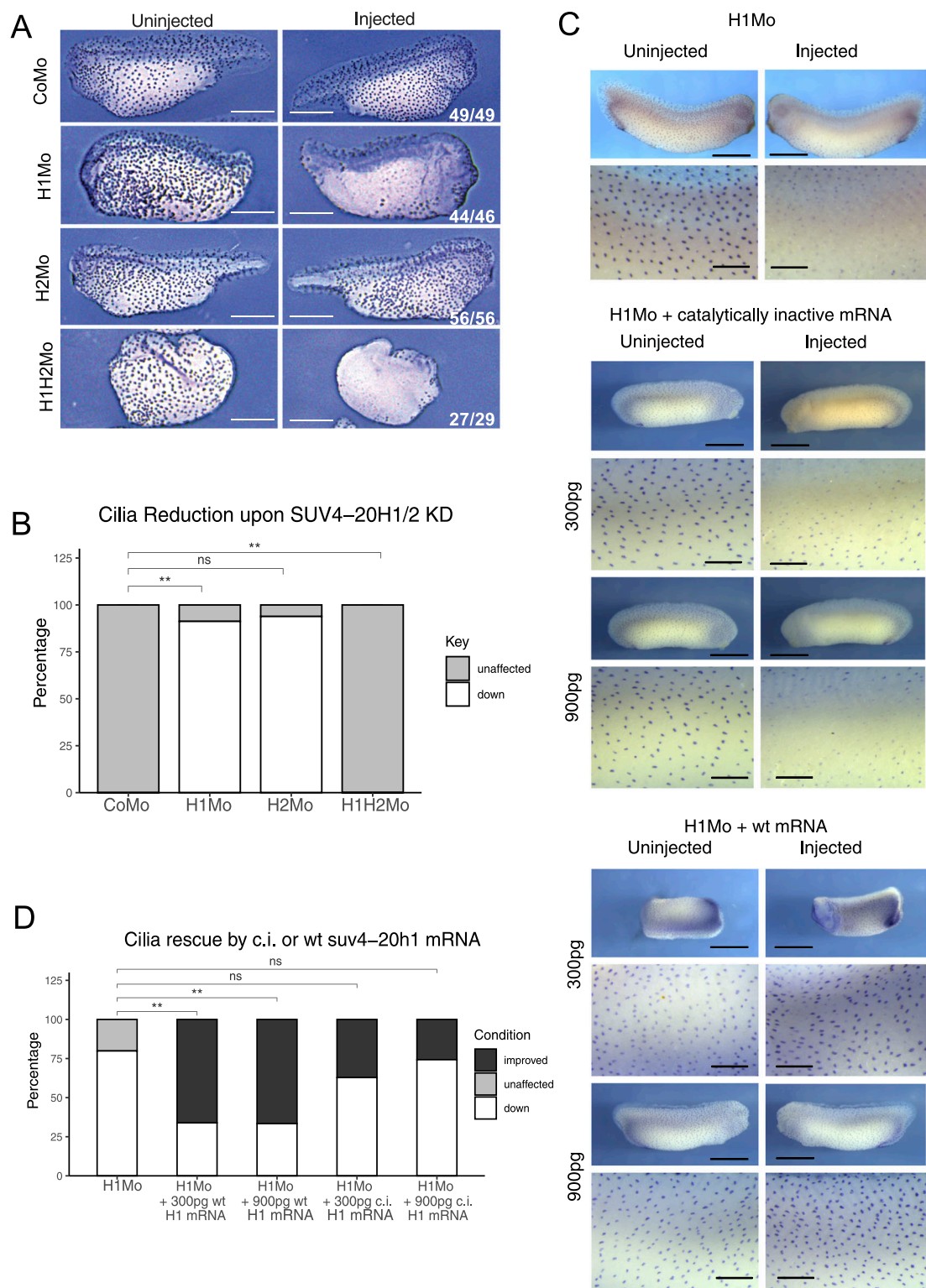

**Figure 5.   SUV4-20H1 activity is needed for cilia tuft formation.**
Representative immunostainings of multiciliated cells (acetylated a-tubulin) on the tail bud stage embryos. **(A)** *X. tropicalis* embryos upon KD of the *suv4-20h* enzymes individually or in concert. Scale bars = 1 mm, n = 3 biol. **(A, B)** replicates (B) Quantification of the experiment in (A). **(C)** *X. laevis* embryos upon KD of *suv4-20h1* and rescued with either catalytically inactive (c.i.) or WT *suv4-20h1* mRNA. Scale bars = 1 mm (whole embryo) and 200 $\mu$m (inserts), n = 3 biological replicates. **(C, D)** Quantification of panel (C).

ciliogenesis in a significant manner, whereas the catalytically inactive variant was indistinguishable from the SUV4-20H1-depleted embryos (Fig 5C and D).

In a separate experimental series, we confirmed that PHF8 significantly restored ciliogenesis in SUV4-20H1 morphant explants by immunostaining (Fig 6A and B). Confocal microscopy confirmed that hPHF8 overexpression improved both formation of the apical actin cap (64% of MCCs) and assembly of ciliary axonemes (75% of MCCs) (Fig 6C–E). In summary, we found no difference in strength and penetrance of the cilia phenotype between single SUV4-20H1 MOs and SUV4-20H1/H2 dMOs. We conclude that the conversion of H4K20me1 to H4K20me2 by SUV-20H1 is essential for the differentiation of MCCs, whereas the contribution from SUV4-20H2 is not required.

# Discussion

The SUV4-20H1/KMT5B histone methyltransferase converts H4K20me1 to H4K20me2 and, along with SUV4-20H2/KMT5C, generates the most abundant histone modification in vertebrate chromatin (Pesavento et al, 2008; Evertts et al, 2013; Jorgensen et al, 2013; Pokrovsky et al, 2021). The work presented here has elucidated an unexpected and novel function for this enzyme in developing Xenopus embryos. On the morphological level, SUV4-20H1 is needed for the production of hundreds of motile cilia on the apical surface of MCCs. These cilia tufts are a hallmark of mucociliary epithelia and generate directional flow along their surface (Boutin & Kodjabachian, 2019). To our knowledge, these data represent the first example of a cellular organelle to depend on a histone-modifying enzyme.

The ciliogenic defect is specific, because it is produced by two independent, non-overlapping suv4-20h1 Mos. It is also significantly rescued by WT, but not enzymatically inactive, SUV4-20H1 protein. Furthermore, axoneme formation is only impaired by SUV4-20H1, which contributes very little to H4K20me3 abundance. Conversely, SUV4-20H2 does not impact ciliogenesis, although it contributes to the global H4K20me2 level. This implies that the SUV4-20h2 enzyme is either not targeted, or has less access, to genes that become misregulated in morphant MCCs. Therefore, the ciliary phenotype represents a specific function of SUV4-20H1 that most likely involves conversion of H4K20me1 to H4K20me2, implying some nonredundant functions for the two enzymes in transcriptional regulation. This is consistent with evidence from knockout mice and FRAP analyses (Schotta et al, 2004; Hahn et al, 2013).

The hypothesis that the cilia phenotype arises from the increase in H4K20me1 levels is independently supported by the return of ciliary axonemes upon PHF8 overexpression. Although this enzyme can remove methyl groups from several sites on histone tails, including H3K9, H3K27, and H4K20, cells depleted for PHF8 predominantly show an increase in H4K20me1 levels at active promoters (Liu et al, 2010). It is important to note that SUV4-20H1 and PHF8 interact very differently with chromatin, that is, PHF8 acts on a limited portion of the genome (mostly promoters), whereas SUV4-20H1 operates broadly. This probably explains why rescue by PHF8

is incomplete, and why we did not detect a reduction in H4K20me1 levels upon PHF8 overexpression in bulk chromatin.

Our data regarding the rescuing activity of PHF8 is interesting in several ways. Because PHF8 exclusively demethylates the monomethylated, but not the di or trimethylated state of H4K20, it corroborates our conclusion that the cilia phenotype is independent from SUV4-20H2 and/or H4K20me3. Second, cilia formation in SUV4-20H morphants is improved even by a truncated Xenopus PHF8 variant, which lacks the c-terminal part with critical amino acid residues of its catalytic domain. This suggests that the proposed repressive effect of hyperabundant H4K20me1 on the transcription of several hundred MCC core genes depends on reader protein(s), whose access could be impaired by the inactive PHF8 variant. Although this remains speculative, members of the L3MBTL protein family could be responsible for mediating transcriptional repression in embryos with high H4K20me1 abundance. Some of these proteins bind H4K20me1 and compact chromatin structure by fostering interactions between neighboring nucleosomes (Trojer et al, 2007; Balakrishnan et al, 2010). It should be noted that besides PHF8, there are additional histone demethylases which recognize methylated H4K20 as substrates (Brejc et al, 2017; Cao et al, 2020). It is currently unknown whether any of these enzymes is involved in MCC differentiation.

Differential gene expression analysis for control and SUV4-20H dMO ACs revealed the concerted down-regulation of about 500 genes associated with the GO terms "cilium" or "cytoskeleton," providing a plausible explanation for the observed phenotype. The expression of 75–80% of these genes was improved by overexpression of hPHF8, consistent with the observed morphological rescue under this condition. However, the total number of both upand down-regulated genes in SUV4-20H1/2 dMOs is much higher, raising important questions: Do all misregulated genes display an increase in the H4K20me1 mark? Can SUV4-20H1 dependent chromatin regions be distinguished from those controlled by SUV4-20H2? Where does H4K20me1 accumulate on the misregulated genes? Finally, are there transcription factor-binding motifs which are common to H4K20me1-sensitive genes? These questions can be addressed in the future by comparative genome profiling of the H4K20me1 landscape in normal and SUV4-20H-depleted ACs.

Our study has firmly established that SUV4-20H1 enzymatic activity is required for the formation of cilia tufts on Xenopus larval epidermis. The most likely explanation for the ciliogenic phenotype in SUV4-20H1 morphants is a transcriptional change because of the altered H4K20me1 landscape. An alternative explanation is suggested by recent findings demonstrating non-histone proteins as substrates for histone methyltransferases (Jambhekar et al, 2019). For instance, the chromatin modifiers SET7, EZH2, and SMYD1 are already known to affect gene expression by methylating specific transcription factors (Jambhekar et al, 2019). SUV4-20H1/KMT5B was shown to methylate some non-histone proteins in vitro (i.e., the zinc finger transcription factor CASZ, and OSBPL1A, a transport protein of oxidized sterols) (Weirich et al, 2016). Although the significance of this finding is currently unclear, it opens up the possibility that the ciliogenic defect could involve mechanisms independent from histone methylation. Although additional experiments are needed to address these questions in the future, our study has firmly

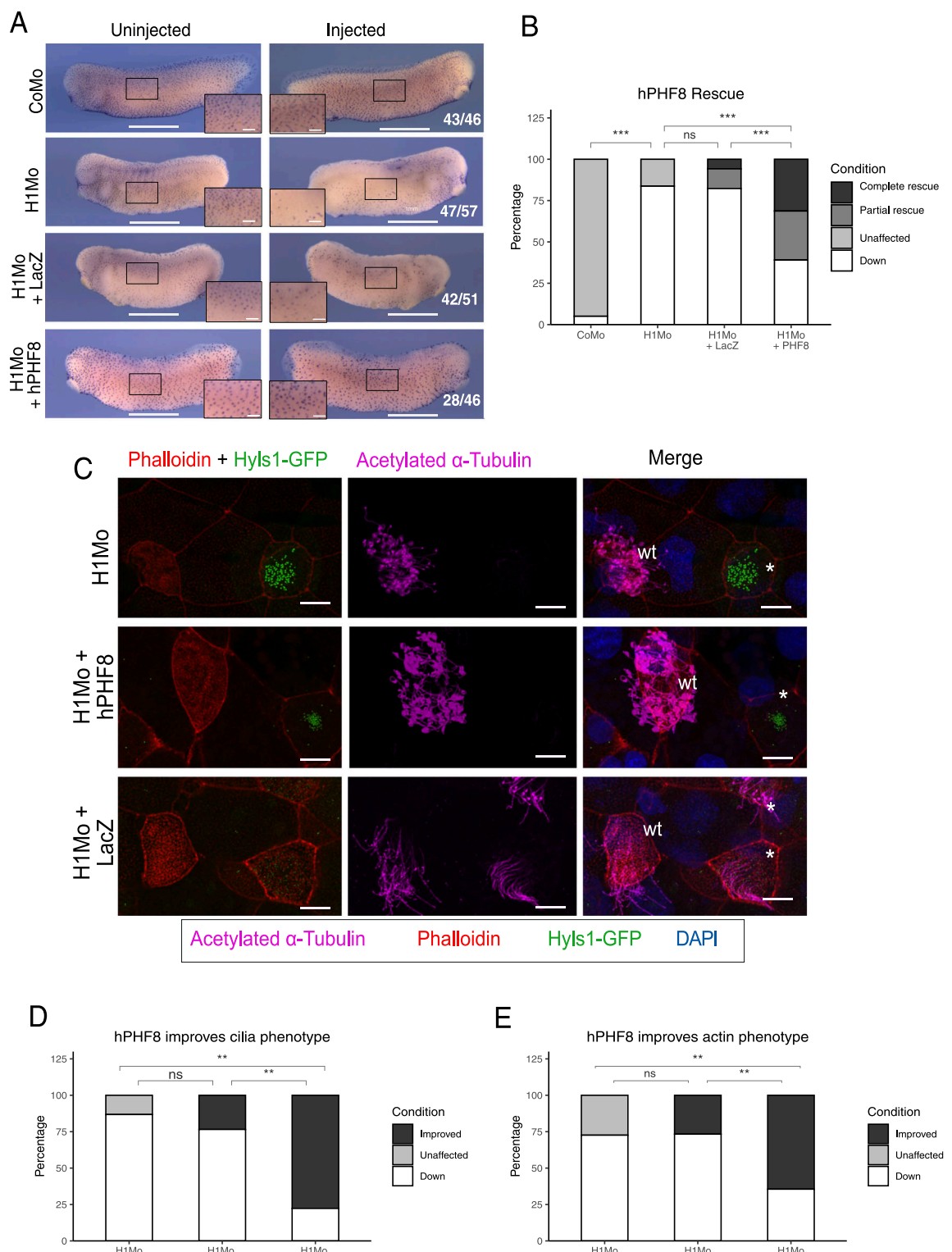

**Figure 6. Rescue of ciliogenic phenotype in H1 Mo embryos with hPHF8.**
**(A)** Representative immunocytochemistry images of the tail bud stage embryos stained for acetylated α-tubulin (multiciliated cells). Injected reagents shown on the left, uninjected, and injected sides shown on top. Scale bars = 1 mm (whole embryo) and 200 μm (inserts), n = 3 biological replicates. **(B)** Quantification of (A). **(C)** Representative confocal images detailing the multiciliated cell phenotype. Injected reagents shown on the left. The basal bodies are green (hyls1-GFP), the ciliary axonemes are magenta (acetylated α-tubulin antibody), the apical actin meshwork is red (phalloidin), and the nucleus is blue (DAPI). A mosaic injection scheme is used allowing KD and wt cells to be present in the same field of view (* = KD MCCs, wt = wildtype MCCs). Scale bar = 10 μm, n = 3 biol. replicates. **(D, E)** Quantification of cilia phenotype (D) and actin phenotype (E).

established the requirement for SUV4-20H1 in the formation of the cilia tuft.

What could be the biological link that connects SUV4-20H1 activity with MCC differentiation? Formation of the cilia tuft requires exit from the cell cycle. We and others have shown that the relative abundance of each H4K20me state largely reflects the cellular proliferation rate. Cell cycle withdrawal establishes a distinct equilibrium, in which un and monomethylated H4K20 marks are reduced by conversion to di and trimethylated states. In consequence, the chromatin of non-proliferating cells switches from primarily mono to primarily dimethylated H4K20, although the low-abundant H4K20me3 mark shows the highest absolute fold increase (Evertts et al, 2013; Schuh et al, 2020; Corvalan & Coller, 2021; Pokrovsky et al, 2021). Consistent with this switch, both SUV4-20H enzymes have been involved in maintenance of quiescence, although with distinguishable and perhaps cell-type-specific contributions (Evertts et al, 2013; Boonsanay et al, 2016). Therefore, we propose that the ciliogenic phenotype is caused by a switch from quiescent H4K20me2-dominated chromatin to a proliferation-like H4K20me1 chromatin that is incompatible with ciliogenic transcription.

MCCs might be particularly vulnerable to such an aberrant chromatin state because their differentiation requires the mitotic oscillator, consisting of CDK1 and APC/C. At high activity levels, the oscillator controls entry and exit of mitosis in proliferating cells, whereas at lower signal levels, it coordinates the progression of organelle remodeling associated with cilia formation (Al Jord et al, 2019). Specifically, postmitotic MCCs redeploy the mitotic oscillator to coordinate centriole amplification with basal body migration and axoneme synthesis (Al Jord et al, 2017), all of which are perturbed in SUV-4-20H-depleted epidermis. Although it is currently unknown how the activity of the mitotic oscillator is calibrated in postmitotic cells, we posit that SUV4-20H1 is needed to relieve cilia genes from the repressive influence of H4K20me1, whose abundance is maximal at the late G1 phase, but virtually absent in the G0 phase. In this model, SUV4-20H1 activity may enable the transcriptome of MCCs to respond adequately to the low activity of the mitotic oscillator as a prerequisite for cilia tuft formation.

We assume that the impact of SUV4-20H activities on cell quiescence and differentiation has remained largely unexplored because most of studies on histone PTMs have been performed in proliferating cells, which are generally undifferentiated. We have detected this mechanism in the highly dynamic context of developing frog embryos, in which cells stop proliferating at specific times and switch on differentiation programs, which let them assemble diverse cytoskeletal structures depending on the cell type. It is important to recognize that the histone modification landscape of cells that undergo this switch is challenged by the sudden disappearance of S-phase dilution of parental histone marks (Scharf et al, 2009). This indeed influences epigenetic information, because histone PTMs are propagated across the cell cycle by at least two distinct kinetic modes (Alabert et al, 2015). We have recently found by computational modeling that dilution through DNA replication is sufficient to explain the abundance of H4K20me states in proliferating embryos, whereas active demethylation is needed to shape histone methylation levels in G1-arrested embryos (Schuh et al, 2020). We therefore believe that for a deeper understanding of histone PTMs and their impact on cellular morphology, it will be necessary to investigate the chromatin landscape in differentiated, non-proliferating cells, which, after all, constitute most of the human body.

On the organismal level, specialized cytoskeletal structures play a role in the etiology of many human diseases. For example, defects in motile cilia formation are accountable for polycystic kidney disease, Meckel–Gruber syndrome or Leber´s congenital amaurosis (Waters & Beales, 2011). Moreover, cilia are essential for the mucociliary clearance in the lung and thus play a pivotal role in several diseases including cystic fibrosis, asthma or chronic bronchitis (Tilley et al, 2015). Our data show that it is possible to manipulate the amount of cilia in both directions by acting on the chromatin enzymes that regulate the abundance of H4K20 methylation. In fact, these enzymes are druggable and small-molecule inhibitors for SUV4-20H enzymes and JmjC-domain-containing proteins like PHF8 are available (Bromberg et al, 2017). We suggest that manipulating H4K20 methylation states could have a therapeutic potential for diseases of unclear etiology that affect tubulin- or actin-based cytoskeleton structures.

# Materials and Methods

### Ethics statement

Xenopus experiments adhere to the protocol on the protection and welfare of animals of the European Commission and are approved by the local animal care authorities.

### Embryo handling and AC preparation

*Xenopus laevis* and *Xenopus tropicalis* embryos were handled and fertilized in vitro using standard procedures. Embryos were injected with up to 10 nl vol. two-cell stage-injected embryos were co-injected with Alexa Fluor-488 Dextran (Invitrogen) as a lineage tracer and sorted by left- or right-side injection before harvesting. To maximize the distribution of reagents, blastomeres at the two-cell stage were injected twice at opposite ends (one cell for half-sided injections, both cells for radial injections). Whole embryos were staged according to the normal table (Nieuwkoop & Faber, 1967). *X. tropicalis* animal caps (ACs) were manually dissected and staged based on sibling embryos.

### Expression constructs and morpholino oligonucleotides

Full-length human *phf8* cDNA in pCMV-SPORT6 and truncated *xphf8* (*xphf8ΔC*) in pCS108 were obtained from Dharmacon/Horizon Discovery. The following published plasmid constructs were kindly provided by: P. Walentek (MCI-hGR), A Schweikert (foxj1, both in pCS2+), and A. Dammermann (hyls1-gfp in pCS2+). Catalytically inactive *suv4-20h1* mRNA was generated by point mutagenesis of *X. tropicalis suv4-20h1* in pCS2+ (N248A and Y283A in NCBI Reference Sequence XP_002941687.1) as described in (Nicetto et al, 2013). Synthetic mRNAs were injected into embryos at the two- or eight-cell stage.

Translation-blocking morpholino oligonucleotides (Mo) directed against Xenopus *suv4-20h1* (complementary to *X. laevis* and *X. tropicalis* suv4-20h1 5′-ggattcgcccaaccacttcatgcca-3′) and *suv4-20h2* (*X. laevis* suv4-20h2: 5′-ttgccgtcaaccgatttgaacccat-3′, *X. tropicalis* suv4-20h2: 5′-ccgtcaagcgatttgaacccatagt-3′) and standard control Mo (5′-cctcttacctcagttacaatttata-3′) were obtained from Gene Tools LLC. *X. laevis* embryos were injected with 30–40 ng of each Mo per blastomere into the animal pole at the two-cell stage. For confocal analysis, *X. laevis* embryos were injected at the eight-cell stage in one dorsal blastomere with 5 ng of each morpholino in 2.5 nl (CoMo = 10 ng). *X. tropicalis* embryos were injected with 20 ng of each morpholino. Rescue experiments were performed by co-injecting *suv4-20h1/2* Mos with mRNAs as specified in figure legends.

### Mass spectrometry analysis and histone PTM quantification

A detailed description of the method and the data analysis has been published (Pokrovsky et al, 2021). In short, for studying histone PTMs, we extracted histones from *X. laevis* embryos at stage NF16 (neurula stage). Absolute and relative abundance of the histone PTMs were measured on the QexactiveHF LC–MS/MS in DDA Top10 acquisition method, using a library of isotopically labelled peptides (R10s), which mimic modification states present on endogenous, tryptic histone peptide fragments, as internal and inter-sample control. The open source Skyline software (version 3.7) was used for the data analysis. Total area MS1 from endogenous peptides was normalized to the respective area of heavy-labeled peptides. The sum of all normalized total area MS1 values of the same isotopically modified peptide in one sample resembled the amount of total peptide. The relative abundance of an observed modified peptide was calculated as the percentage of the overall peptide.

The spectra from the samples, analyzed in Fig 1, were missing the signal from the H4K20un heavy-labeled peptide. H4K20me1, me2, and me3 heavy-labelled peptides were detected as expected. The missing values (12% of the entire dataset) were imputed using stochastic regression imputation method before calculating the relative abundance. In short, for the imputation step, the ratios between R10 H4K20un to me1, me2, and me3 were calculated based on more than 50 previous MS experiments, where the same R10 library was used. Based on the ratios, the H4K20un R10 values were calculated, averaged, and then a random mean was chosen for the relative abundance calculation.

### Immunocytochemistry

ICC was performed as described (Robinson & Guille, 1999). For confocal analysis, the methanol step was omitted. Embryos were stained with: monoclonal anti-acetylated tubulin antibody Sigma-Aldrich (T6793, diluted 1:500), 25 $\mu$M DAPI, 0.33 $\mu$M phalloidin-Alexa 555 Thermo Fisher Scientific (A34055). Depending on the analysis, embryos were incubated with secondary goat anti-mouse conjugated to Alexa 647 (A21236; diluted 1:500; Thermo Fisher Scientific) or alkaline phosphatase-fused secondary sheep anti-mouse antibodies (AP303A; diluted 1:1,000; Chemicon).

### Statistical analysis

ICC results from knockdown experiments were analyzed using a two-tailed *t* test. Results from rescue experiments were analyzed using one-way ANOVA with post-hoc Tukey test.

### RNA library preparation and sequencing

Total cellular RNA was isolated and purified from approximately 30 *X. tropicalis* ACs using TRIzol (Ambion) and phenol/chloroform extraction, followed by clean-up with RNeasy Mini-Kit (QIAGEN). RNA Integrity Number (RIN) was analysed using an Agilent 2100 Bioanalyzer to monitor RNA quality (Schroeder et al, 2006). Ribosomal RNA was removed from 500 ng of input RNA using either Ribo-Zero Gold rRNA Removal Kit (Human/Mouse/Rat) from Illumina or NEBNext rRNA Depletion Kit (Human/Mouse/Rat). Then, total stranded RNA sequencing libraries were prepared using NEBNext Ultra Directional RNA Library Prep Kit for Illumina following the manufacturer's instructions. The quality and size of libraries were verified using the Agilent Bioanalyzer with the Agilent DNA 100 kit. RNA libraries were multiplexed and sequenced with 50 base pairs (bp), paired-end reads to a depth of 30 million reads per sample on an Illumina HiSeq4000.

### RNA-seq analysis

Sequencing reads were mapped to the reference genome (*X. tropicalis* v9.1 available from Xenbase) using STAR (version 2.7.1a). Gene models were provided as a gtf file, which was converted from the Xenbase gff3 format using Cufflinks gffread (version 2.2.1). Reads were counted for each gene in the same STAR run by the quantMode GeneCounts.

Downstream analyses were carried out in R (version 3.6.1) using helper function from the HelpersforDESeq2 package (version 0.1; https://github.com/tschauer/HelpersforDESeq2) Differential analysis was performed by DESeq2 (version 1.26) for each pairwise comparison separately. Genes with at least 1 mapped read detected in 75 percent of the samples in the given dataset were considered. Significantly differential genes were defined by an adjusted *P*-value cutoff of 0.05. Results were visualized as MA plots, where the x-axis indicates the $\log_{10}$ mean counts and the y-axis the $\log_2$ fold change ($\log_2$FC). Comparison of the independent datasets was visualized by $\log_2$FC–$\log_2$FC plots, where the shown condition was compared with its own control.

Gene ontology annotation was derived from the mouse org.Mm. eg.db package (version 3.8.2) by converting gene ids from mouse to Xenopus. Genes annotated with cellular component "cilium" or "cytoskeleton" were selected by the GO ids "GO:0005929" or "GO:0005856" and all their offspring terms, respectively. GO enrichment analysis was performed by the topGO package (version 2.36.0) using Fisher statistics. GO results were visualized as bubble plots, where the x-axis indicates the fold enrichment (i.e., the observed number of significant genes for the GO term over the expected number), the bubble size is proportional to the number of significant genes for the GO term and the color intensity is related to the Fisher test *P*-value.

## Data Availability

RNA high-throughput sequencing data have been deposited in the NCBI GEO under accession number GSE161251.

## Supplementary Information

## Acknowledgements

We thank Barbara Hölscher for exceptional embryo stainings; Drs. Hiroshi Kimura, Chris Kintner, Gunnar Schotta, Axel Schweickert, Alexander Dammermann, and Peter Walentek for their kind gift of antibodies and recombinant plasmids; Dr. Andreas Thomae and the Core Facility of Bioimaging of the Biomedical Center, LMU Munich for instructions and technical support in confocal imaging, and the BMC Core Facility Animal Models (CAM) for animal care and maintenance. This work was funded by the Deutsche Forschungsgemeinschaft (DFG, German Research Foundation)—Project-ID 213249687—SFB 1064 (Project A12).

### Author Contributions

A Angerilli: validation, investigation, visualization, methodology, and writing—original draft.
J Tait: validation, investigation, visualization, methodology, and writing—original draft, review, and editing.
J Berges: investigation and methodology.
I Shcherbakova: investigation.
D Pokrovsky: investigation and visualization.
T Schauer: data curation, software, formal analysis, and visualization.
P Smialowski: data curation, software, and formal analysis.
O Hsam: investigation.
E Mentele: investigation.
D Nicetto: investigation and methodology.
RAW Rupp: conceptualization, resources, supervision, funding acquisition, investigation, methodology, project administration, and writing—original draft, review, and editing.

### Conflict of Interest Statement

The authors declare that they have no conflict of interest.

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
