## [Reviewer comments · Life Science Alliance]

Life Science Alliance

Histone methyltransferase SUV4-20H1/KMT5b is required for *Xenopus* multiciliated cell differentiation

Alessandro Angerilli, Janet Tait, Julian Berges, Irina Shcherbakova, Daniil Pokrovskii, Tamas Schauer, Pawel Smialowski, Ohnmar Hsam, Edith Mentele, Dario Nicetto and Ralph Rupp

DOI: <https://doi.org/10.26508/lsa.202302023>

Corresponding author(s): Prof. Ralph A.W. Rupp (Ludwig-Maximilians-Universität München)

Review Timeline:

Submission Date:	2023-03-06
Editorial Decision:	2023-03-07
Revision Received:	2023-04-18
Accepted:	2023-04-18

Transaction Report:

Please note that the manuscript was reviewed at Review Commons and these reports were taken into account in the decision-making process at Life Science Alliance.

Review
COMMONS

Manuscript number: RC-2021-00743

Corresponding author(s): Ralph A.W. Rupp

1. General Statements [optional]

To the Editors of the Journal of Cell Biology,

In our revised manuscript we reveal a novel function for SUV4-20H1, one of two enzymes that generate H4K20me2, the most abundant chromatin modification in vertebrates. Employing *Xenopus* embryos as model system, we provide consistent and robust evidence that the enzymatic activity of SUV4-20H1/KMT5B is indispensable for the formation of motile cilia tufts on the apical surface of multiciliated cells, a key component of mucociliary epithelia. Consistent with the observed phenotypic rescue by PHF8 overexpression, we conclude that the conversion of the H4K20me1 to H4K20me2 state by SUV4-20h1 is needed for cilia tuft formation. In contrast, the related SUV4-20H2 enzyme that catalyzes H4K20me2/me3 states, is not involved in ciliogenesis.

As shown by the attached, marked up copy of the manuscript, we have thoroughly revised title, main text and figure legends and included new experimental data to address the reviewers' comments. This new data includes quantitative mass spectrometry to reveal the individual and combined consequences of the SUV4-20H enzymes on the H4K20me landscape in the embryo (Fig. 1). We also show that the enzymatic activity of SUV4-20H1 is specifically needed for cilia tuft formation (Fig. 5). The statistical overlap between the GO terms "cytoskeleton" and "cilium" is presented in Figure S3. MA-Plots to visualize transcriptomes of control conditions are provided in Figure S5.

In addition, we have omitted Figure 7, which summarized our findings in a speculative model that put our findings in context with known knowledge about H4K20 methylation in the course of the cell cycle. We acknowledge in retrospect that this model was too speculative, since we have not studied the cell cycle phases of SUV4-20H depleted multiciliated cells.

Two experiments, suggested by reviewers were not addressed during the revision. First, we have not manipulated PRSET7/KMT5A in order to test, whether the accompanying reduction in H4K20me1 abundance could improve the ciliogenic defect. PRSET7 generates H4K20me1 on newly synthesized histones during G2/M phases. Loss of PRSET7 is early embryonic lethal in mice, and it is known that insufficient amounts of the H4K20me1 mark on chromatin leads to severe defects and arrest in mitosis. We therefore consider the suggested experiment as not informative, since cells arrested in mitosis will not be able to differentiate and thus cannot shed light on the ciliogenic defect of multiciliated cells in G0-phase. Second, we consider the request to perform additional ChIP-seq analysis as being out of the scope of this manuscript. Our main claim and novel result that SUV4-20H1 is needed for ciliogenesis in the mucociliary epithelium of the larval skin of *Xenopus*, is justified by the presented experimental data and does not depend on ChIP-seq analysis.

Full Revision

We would like to thank the reviewers at *Review Commons* for their time, expertise and constructive criticism and hope that our point-by-point response, which details the improvements we have implemented in the revision, make our manuscript attractive for publication in the *Journal of Cell Biology*.

With best regards,

Ralph Rupp
on behalf of all co-authors

In blue - reviewers' comments

In black - our response

Reviewer #1

Summary

In this manuscript, the authors describe a novel function of the lysine methyltransferase SUV4-20H1 and SUV4-20H2 enzymes in *Xenopus* embryos. They show that the depletion of SUV4-20h1, SUV4-20H2 or both enzymes in frog embryo via antisense Morpholino oligonucleotides impairs the di and tri-methylation states of the lysine 20 of histone H4 (H4K20) leading to an improper enrichment in mono-methyl H4K20 (H4K20me1). This was associated with alterations in the differentiation of multiciliated cells in the frog larval epidermis, as observed by the strong reduction of cilia numbers in these cells. The authors show that these defects in ciliogenesis are related to perturbation of cytoplasmic processes, likely caused by a down-regulation in the expression of cytoskeleton and cilium related genes upon depletion of SUV4-20H enzymes. Strikingly, the depletion of SUV4-20H1, but not of SUV4-20H2, appears sufficient to induce these differentiation defects, which can be rescued by the overexpression of the human H4K20me1 demethylase PHF8 or a truncated form of *Xenopus* PHF8 potentially compromised in its catalytic activity. Since the overexpression of PHF8 proteins can improve the expression levels for the majority of cytoskeleton and cilium genes downregulated upon SUV4-20H1/H2 depletion and that PHF8 targets H4K20me1, the authors propose the conversion of H4K20me1 to higher K4K20me states by SUV4-20H1 is critical for proper expression of these genes and the formation of cilia during multiciliated cells differentiation in *Xenopus* embryos.

Evidence, reproductibility and clarity

Although the description of a function of SUV4-20H enzymes in ciliogenesis is particularly interesting and novel, the results do not seem convincing enough on the understanding of the mechanisms involved. Several conclusions seem a bit speculative, especially concerning the proposed critical role of H4K20me1 to H4K20me2 conversion in the control of cytoskeleton and

cilia gene expression. As described below, several results appear contradictory and further experiments and controls are necessary before such a possibility can be proposed.

(i) In Figure 1A, the depletion of SUV4-20H2 triggers the same reduction in H4K20me2 and increase in H4K20me1 than the depletion of SUV4-20H1. Yet, in contrast, depletion of SUV4-20H2 does not seem to induce the loss of cilia (Figure 5A). Therefore, how to conclude that the alterations in differentiation of multiciliated cells described here are related to defects in the conversion of H4K20me1 to H4K20me2 ?

We thank the reviewer for raising this point. The simple answer is that although SUV4-20H1/2 enzymes catalyze the same reactions *in vitro*, it is undisputed that they have at least partly non-overlapping functions *in vivo*:

- First, intercrosses of *suV4-20h1* +/- mice produce *suV4-20h1* -/- pups at sub-mendelian ratios, indicating this enzyme to be essential for development. In contrast, *suV4-20h2* null mice are fully viable (Schotta et al., G&D 2008, pmid: 18676810).
- Second, the analysis of H4K20 methylation states in bulk chromatin of primary MEFs with single and double-null genetic status has indicated that SUV4-20H1 is largely responsible for di-methylated, but not tri-methylated H4K20, whereas SUV4-20H2 has a mild effect on the dimethylated state, but catalyzes predominantly H4K20me3 (*ibid.*).
- Third, FRAP analysis in MEFs revealed that SUV4-20H2 protein stably associates with heterochromatic areas, while SUV4-20H1 interacts in a highly dynamic manner with chromatin and is not enriched in heterochromatin (Hahn M. et al., G&D 2013, pmid: 23599346).

To define the individual contributions of each enzyme to the H4K20 methyl landscape in *Xenopus* embryos we have performed **new experiments**, in which we accurately quantified the four H4K20 methylation states (me0, me1, me2, me3) for wt, control morphant, single and double morphant conditions by quantitative mass spectrometry. The results are compatible with the Western blot data, but provide a much more accurate measurement of the abundance of each H4K20 methyl state in an antibody independent manner. **This new data replaces the western blot analysis in Fig. 1A:**

Relative abundance of H4K20 methylation states in wild-type (wt), control morpholino (CoMo), SuvH1 morpholino (H1), SuvH2 morpholino (H2), and double SuvH1/SuvH2 morpholino (H1H2) injected embryos at developmental stage NF17. N≥3 biological replicates; mean ± s.e.m. "p" – propionylated (= me0, naturally unmodified). Note different y-axis scale for the me3-state in panel B.

The mass spec data indicates that both SUV4-20H enzymes contribute about equally to the global H4K20me2 levels in *X. laevis*, while SUV4-20H1 has a negligible impact on H4K20me3 levels that are dependent on SUV4-20H2. Overall, this matches well with the findings from murine MEF cells of knockout mice, except for the higher impact of SUV4-20H2 on H4K20me2 in frogs, which might be due to the different experimental systems (immediate response to protein knockdown in F0 embryos versus potentially compensated H4K20me2 levels in cultured MEF cell lines from SUV4-20H2 *-/-* mice). Due to the lack of specific antibodies for these enzymes, it is not possible to directly address the endogenous enzyme levels or their genome-wide chromatin profiles.

In light of the undisputed mouse data, our key result that only SUV4-20H1 is required for multiciliogenesis implies that the SUV4-20h2 enzyme is either not targeted, or has less access, to genes that become misregulated in morphant MCCs. We have added this notion to the second paragraph found on page 12.

(ii) It is intriguing that a truncated form of *Xenopus* PHF8 (potentially compromised in its catalytic activity as suggested by authors) can improve the expression levels for the majority of cytoskeleton and cilium genes downregulated upon SUV4-20H1/H2 depletion (Figures 3, 4, S4; S5 etc.). Does it mean that this rescue is not dependent on the demethylase activity of PHF8? Actually, it is not indicated (by immunoblot or microscopy) whether overexpression of hPHF8 and xPHF8 proteins can indeed decrease the levels of H4K20me1 in frog embryos upon SUV4-20H depletion. This control is absolutely required. Indeed, PHF8 has multiple targets and it is not specific to H4K20me1. If changes in cytoskeleton and cilium genes expression are indeed related to improper levels of H4K20me1 on chromatin, the depletion of the H4K20me1 methyltransferase PR-set7 should rescue their expression. A such experiment might worth it to be performed. Another critical approach will be to evaluate by ChIP the levels of H4K20me1 and H4K20me2 at these genes under *Suv4-20h* dMO conditions and PHF8 overexpression.

We appreciate the constructive comments of the reviewer, aimed to help us find orthogonal means to improve the impact of our data.

Based on the available structural data, the truncated *Xenopus* PHF8 variant is predicted to be enzymatically inactive. We assume its rescuing activity to represent a dominant-negative interference effect, whereby the inactive PHF8 blocks binding of endogenous proteins known to recognize the H4K20me1 mark. These include so far HCF-1 (Julien and Herr, *Mol Cell* 2004, PMID: 15200950), 53BP1 (Houston et al., *JBC* 2008, PMID: 18480059), MSL3 (Kim et al., *Nat Struct Mol Biol.*2010, PMID: 20657587), L3MBTL1 (Trojer et al., *Cell* 2007, PMID: 17540172; Sakaguchi et al., *PONE* 2012, PMID: 23024815) and members of the condensin II complex such as CAP-G3, CAP-G2 (Centore et al., *Mol Cell* 2010, PMID: 20932472; Liu et al., *Nature* 2010, PMID: 20622854).

It is not clear, whether overexpression of PHF8 can be expected to change H4K20me1 abundance on the bulk chromatin level. In proliferating cells, SET8/KMT5a monomethylates during G2/M-phase most of the newly incorporated H4 proteins that are de novo synthesized in the preceding S1-phase (i.e. 50% of all H4 proteins in the nucleus). Within two to three rounds of

replication, about 90% of new H4 histones become monomethylated at Lysine 20 (Pesavento et al., Mol Cell Biol 2008, pmid: 17967882). Consequently, new H4K20me1 marks are expected everywhere in chromatin (for review see Jorgenson S. et al., Nucl Acids Res 2013, pmid: 23345616), in particular under SUV4-20H enzyme depletion conditions, which prevent its conversion to H4K20me2 significantly. In contrast it is known that PHF8 acts near transcriptional start sites (Qi et al., Nature 2010, pmid: 20622853; Liu et al., Nature 2010, pmid: 20622854; Asensio-Juan et al., Nucl Acids Res 2012, pmid: 22850774), and thus acts on a significantly smaller fraction of chromatin.

Despite these doubts, we have performed new experiments to measure the abundance of all four H4K20 methyl states in double morphant, PHF8 overexpressing, and PHF8-rescued embryos:

Relative abundance of H4K20 methylation states in wild-type (wt), SUV4-20H1/SUV4-20H2 double morpholino (H1H2), PHF8 overexpressing (PHF8), and PHF8-rescued double-morphant embryos (Rescue300 and Rescue900) at developmental stage NF17. Single phf8 mRNA injections were done with 900pg/embryo; in rescue experiments, phf8 mRNA was injected at 300pg/embryos and 900pg/embryo. N biological replicates are indicated in the brackets for each experimental condition; mean \pm s.e.m. “p” – propionylated (= me0, naturally unmodified). Note different y-axis scale for the me3-state in panel B.

H4K20me1 levels were not reduced by wtPHF8 (see “p”- and “me1”-columns). The marginal increase in H4K20me2/me3 marks, observed in rescued compared to double morphant samples, is not statistically significant. We also noted no difference for histone H3 methyl states on histone H3 Lysine 9 and 27 in bulk chromatin. Altogether, this points to a local effect of overexpressed PHF8, which remains masked in genome-wide elevated levels of H4K20me1.

The proposed PR-SET7 knockdown experiment suffers from several complications one cannot ignore. First, the ciliogenic defect in SUV4-20H morphants results from a transcriptional misregulation in post-proliferative MCCs. Such cells are expected to have extremely low levels of PR-SET7, because this enzyme is actively degraded in G1-phase (see Abbas et al., 2010, pmid: 20932471). Second, PR-Set7 is an essential gene in mice, whose absence results in aborted development already around 2.5 dpc (Huen et al., 2008, pmid: 18319261; Oda et al., 2009, pmid: 19223465). Therefore, we do not consider PR-SET7-knockdown as a possibility to reduce H4K20me1 levels at frog neurula stages without massive side effects on embryogenesis.

Minor comments:

- Figure 1A, why coMO (control ? not described in figure legends) induces higher levels of H4K20me3 and a decrease in H4K20me2 ? please provide quantification of the immunoblots. How long after morpholino injection do we observe changes in H4K20me levels ?

“CoMo” stands for unrelated Control Morpholino, as now explained at its first mentioning on page 5, top paragraph. Our new mass spec data confirms that CoMo injections have no effect on H4K20 methyl states (see **new Figure 1A**).

SUV4-20H morpholino injections affect gene expression in *Xenopus* neuroectoderm from mid gastrula (stage NF11), and change H4K20 methyl abundance at least until tadpole stages (NF33; see Nicetto et al., PLoS Genetics 2013, pmid: 23382689). The ciliogenic defect reported here also persists at least until tadpole stages. Thus, the experimental alteration of H4K20 methyl states coincides with the ciliogenic phenotype.

- in Figure S2A, authors conclude that SUV4-20H regulate multiciliogenesis independently of MCI and FOXJ1 (page 6). Yet, they cannot exclude that SUV4-20H is a downstream essential co-factor of MCI and FOXJ1 proteins.

We agree with this statement and have changed our text accordingly (Page 6).

- what are the H4K20me1/2/3 levels at different timepoints (gastrula and neural) of RNA seq analysis ? do *su4-20h* dMO sufficient to induce changes in H4K20me1/2/3 levels at gastrula stage ? and if it is indeed the case, why cytoskeleton and cilium gene expression are not affected at this stage ?

- For another project we have monitored H4K20me0/1/2/3 levels by quantitative mass spectrometry in the course of development. As shown in the Figure below, me2 and me3 levels increase gradually over time, at the expense of me0 and me1 abundance (Pokrovsky et al., PLoS Biology 2021, pmid: 34491983):

Relative abundance of H4K20 methylation states in wild-type embryos at indicated developmental stages. N=3 biological replicates; mean \pm s.e.m. “p” – propionylated (= me0, naturally unmodified). Note different y-axis scale for the me3-state in panel B.

Full Revision

We have not determined from when on H4K20 methyl states are altered by SUV4-20h Morpholinos, but - as pointed out above - changes in gene expression are observed from mid gastrula stage onwards, i.e. at the same time when MCC precursors become determined in prospective epidermis (Walentek & Quigley, Genesis 2017, pmid: 28095645). Importantly, these events are dictated by temporally fixed developmental programs. The differentiation of MCCs relies on the concerted activation of several hundred “core MCC genes”, which become activated by the transcriptional regulator MCIDAS, which itself is transcriptionally induced in MCC precursors during gastrulation (ibid.). Therefore, ciliogenic mRNA levels cannot be assessed before the MCC differentiation program has been initiated through MCIDAS/FOXJ1.

- in the discussion, the authors indicate that the phenotype is rescued by injecting morpholino-insensitive *su4-20h1/2* mRNAs (not shown). Does the cilia phenotype depend on *Suv4-20h* activity? In other words, does the expression of a morpholino-resistant catalytic dead version of *su4-20h1* rescue or not the cilia phenotype? This is not trivial since SUV4-20h proteins may have functions independent of their methylase activity.

In the revised manuscript, we have tested a catalytically dead SUV4-20h1 point mutant (Nicetto et al., PLoS Genetics 2013, pmid: 23382689) and found that this variant does not rescue the ciliogenic defect of SUV4-20H1 depleted embryos or epidermal organoids. This **new data is found in Figure 5**. Therefore, catalytic activity of SUV4-20H1, which converts H4K20me1 to H4K20me2 state, is required for normal ciliogenesis.

- In discussion, authors claim that unmethylated H4K20 is neutral in terms of transcription? Please provide the reference for such a conclusion. To my knowledge, there is no data in the literature supporting this possibility.

We have removed this statement.

- the title of the paper seems to me to be a bit inappropriate. What link is there with the phases of the cell cycle, no experiment in the manuscript addresses this problem.

We have changed the title of the manuscript so that it exclusively refers to the data and findings we present.

- please revise the text of figure legends. Some look incomplete, confuse or too succinct. See my comment on Figure 1A.

We have thoroughly revised all Figure legends.

Reviewer #1 (Significance (Required)):

Cilia is an ubiquitous organelle that protrudes from the surfaces of many cells, and whose architecture is highly conserved during evolution. Due to the importance of these cell structures, defects in ciliogenesis can lead to numerous human diseases. Using *Xenopus* embryo development as a model system, this study describes for the first time a regulatory role of the lysine methyltransferase SUV4-20H1 in the regulation of the genetic program of multiciliogenesis and how this function is essential for the proper final differentiation of multiciliated cells. It also

Full Revision

indicates how targeting SUV4-20h1 might be an interesting way to modulate ciliogenesis in human-related disease, if this function is evolutionary conserved. To date, the role and regulation of Suv4-20h1 remains poorly understood and this study is clearly attractive in this context. The hypothesis that the conversion of H4K20me1 to H4K20me2 could be important for the regulation of gene expression sounds also particularly interesting. However, at this stage, this clearly requires additional experiments to be fully demonstrated.

This manuscript might be an interest for scientific communities working on ciliogenesis or Suv4-20h enzymes.

We thank the reviewer for acknowledging the novelty of our findings and their potential impact on ciliopathies. With the new experiments (mass spec data, catalytic dead SUV4-20H1) our revised manuscript demonstrates convincingly that the ciliogenic phenotype arises from deficient SUV4-20H1 catalytic activity. This finding is completely unexpected and will certainly attract the interest of both epigeneticists and developmental biologists.

Reviewer #2 (Evidence, reproducibility and clarity (Required)):

In this paper Angerili et al describe new findings suggesting that Suv4-20h1 plays an important regulatory role in ciliogenesis in *Xenopus* embryos.

Investigating embryos with *suv4-20h1* and *suv4-20h2* knockdown and double knockdowns, the authors present convincing phenotypic data that these enzymes are critical for ciliogenic genes expression in ectodermal explants. Inactivation of Suv4-20h1 alone is sufficient for cilia loss. Importantly, the observed phenotype can be rescued by overexpressing Phf8, an H4K20 demethylase. Since *suv4-20h1* and *suv4-20h2* are responsible for H4K20 di and trimethylation, the authors conclude that conversion of H4K20Me1 to H4K20Me2 is essential for the differentiation of multiciliated cells, while up-methylation to H4K20Me3 is not.

The findings are novel and contribute to our understanding of the role of chromatin modifiers in ciliogenic gene expression program. The work is interesting and the main conclusions are supported by the phenotypic and transcriptomic data. The data indirectly indicate that H4K20Me1 represses transcription of cilium and cytoskeleton-associated genes. Although, direct evidence is missing, the paper presents an interesting model suggesting that H4K20Me1 accumulation in G2 phase cells may keep cytoskeletal gene expression under control to prevent interference with centrosome function or the assembly of the mitotic spindle.

The paper could be improved by providing H4K20 methylation mapping data, as they may determine the positions of the modified nucleosomes (coding regions or promoters/enhancers) and the true correlation between the H4K20 mono-di and trimethylation and gene activity.

We thank the reviewer for insightful comments and for acknowledging the novelty and importance of our findings. To the best of our knowledge, this is the first description of a histone modifying enzyme controlling the formation of a specialized cellular organelle.

Full Revision

In keeping with the reviewer's opinion that we have presented "convincing phenotypic data" and that "the main conclusions are supported by the phenotypic and transcriptomic data", we are convinced that additional ChIP-seq analyses aimed to map the areas, whose H4K20 methyl landscape is under control of SUV4-20H and PHF8 enzymes, are beyond the scope of this manuscript.

****Minor points:****

1. Many histone modifiers have also a number of physiologically relevant non-histone protein substrates. The scenario of *suv4-20h1* acting through methylation of regulatory proteins other than histones should be discussed.

This advice is well taken. In our revised manuscript we have discussed this possibility extensively starting in the last paragraph on page 13 (Discussion).

2. It is more appropriate to use the official nomenclatures *KMT5B* and *KMT5C* instead of *suv4-20h1* and *suv4-20h2*.

In our revised manuscript we introduce properly the official names of the enzymes at first mentioning, starting with the title. For the sake of many colleagues, which still connect published information to the old enzyme names, we chose to keep the latter names in addition.

Reviewer #2 (Significance (Required)):

The findings are novel and important.

1 There is some "perceived" controversy about the role of H4K20 methylation in the literature. In some cases (as also suggested in this paper), H4K20me1 has been described as inhibitory for transcription. On the other hand actively transcribed gene body nucleosomes in all cell types are monomethylated at H4K20. If the authors could provide ChIP-seq data for H4K20 methylations, the contribution of this study to the field would be much greater.

We share the reviewer's interest in general aspects of H4K20me1 dependent transcriptional regulation. Our findings for ciliogenic genes are not in conflict with studies reporting the presence of H4K20me1 as a more or less general feature of active genes (e.g. Talaz et al., JBC 2005, pmid:16166085; Vakoc et al., Mol Cell Biol 2006, pmid: 17030614; Barski et al., Cell 2007, pmid: 17512414). Notably, its function there has not been experimentally investigated and the colocalization of this mark with transcriptionally engaged chromatin could have many reasons. On the other hand, short-term inhibition of PRSET7/KMT5A (Congdon et al., J Cell Biol 2010, pmid: 20512922) or PHF8/KDM7B (Liu et al., 2010; Qi et al., 2010; Asensio-Juan et al., 2017) has provided irrefutable evidence for either repressive or activating influences on gene transcription by H4K20me1. A common feature of all these studies, including ours, is that H4K20me1 function is context-dependent.

In the revised manuscript this topic is introduced in the chapter "The histone demethylase PHF8/KDM7B improves ciliogenesis in SUV4-20H dMO epidermis" (RESULTS, page 8) and

Full Revision

major references are provided. It should be noted that the new experimental data with the enzymatically inactive SUV4-20H1 variant (see revised Figure 5) that cannot rescue cilia in *suv4-20H1* MOs, strongly supports the conclusion that converting H4K20me1 to H4K20me2 by SUV4-20H1 is a prerequisite for appropriate cilium gene expression levels.

Reviewer #3 (Evidence, reproducibility and clarity (Required)):

****Summary:****

The authors demonstrate the role of Histone H4K20 methylation in multiciliated cell differentiation using a *Xenopus* model. They show that methylation state is cell cycle dependent, and influences gene expression related to cytoskeletal components and dynamics over time. Overall their manuscript is well-written and their data convincing. As such I have only Minor Comments, below.

****Major Comments:****

(None)

****Minor Comments:****

- The authors detected differential gene expression with changes in H4K20 methylation state using a variety of methods. It would be interesting to take this a step further and explore genomic deposition of H4K20 via ChIP-Seq to determine if this correlates with observed patterns of gene expressions between the various experimental groups as reported.

We thank the reviewer for the positive evaluation! As outlined in our comments to reviewer 1 and 2, our main finding of the enzymatic activity of SUV4-20H1/KDM5B being essential for the formation of cilium tufts on the larval epidermis of *Xenopus* is sufficiently supported and does not depend on additional ChIP-seq analysis.

- In Figure 1C and 1F, it appears there are two segments of the CoMo bar. What do these represent? The way the bar graphs are condensed makes it difficult to appreciate these groups as distinct.

In Figure 1C and G the two segments of the CoMo bar are “down” in white, and “unaffected” in grey, as indicated in the legend. We recognize that the size and dimensions of the columns make this difficult to read. We have revised the design of the figure so that these groups are more distinct.

- Can the authors postulate why there was such a large group of unchanged gene expressions in the H1H2Mo vs CoMo conditions (Figure 2A)?

We propose that genes which respond to the SUV4-20H knockdown are involved either in cell cycle progression and/or in ciliogenesis and may be coupled to the mitotic oscillator, which regulates both processes (Al Jord, Spassky and Meunier, Biol Cell 2019; pmid: 30905068). In contrast, several other studies reported that H4K20me1 is located on many, if not all, actively transcribed genes (Talaz et al., JBC 2005, pmid: 16166085; Barski et al., Cell 2007, pmid: 17512414; Vakoc et al., Mol Cell Biol 2006, pmid: 17030614). Transcription of some of these genes is suppressed by this mark (Congdon et al., J Cell Biochem 2010, pmid: 20512922).

Taking these findings together, it seems plausible that active genes, which maintain the H4K20me1 mark in the presence of SUV4-20h enzymes, will remain suppressed in SUV4-20H depleted chromatin and thus show an unchanged expression. This implies a mechanistic difference between responding and non-responding genes, which may involve targeted activities of H4K20-specific demethylases such as PHF8.

- Can the authors comment on a potential mechanism for the downregulated genes (H1H2Mo vs CoMo set) that went further down with hPHF8 rescue (Fig 4 B, C E)

At the current point, we consider these to represent indirect effects arising from the transcriptional demands of a differentiating cell type in G0-phase being confronted with an H4K20 methyl landscape typically found in proliferating cells. This is a great question that will be interesting to address in the future.

- What was the overlap of the set of genes in the cytoskeleton and cilia groups? Was it significant?

The requested information is found on page 8, top paragraph and in the **new supplementary Figure S3A and B**.

The GO terms “Cytoskeleton” and “Cilia” are not exclusive, but biologically related terms. Therefore, there are overlapping genes between the two groups. Numbers of overlapping genes are shown for all of the genes in the dataset (A), and only the significantly downregulated genes (B). The overlaps are significant for both gene sets.

- What explanation may be plausible for the results quantified in figure 5B, where it appears there was less cilia in the H1Mo group than the H1H2Mo group?

There is no significant difference between the percentage of affected embryos in the H1Mo group compared to the H1H2Mo group (Student's T-test $p = 0.73$).

Reviewer #3 (Significance (Required)):

****Significance:****

The authors present a novel characterization of H4K20 methylation states in the context of cell

Full Revision

cycle and ciliary production / function in a *Xenopus* model. This work sheds light on the role of various methylation states throughout the cell cycle and its effect on gene expression. These findings may be of interest to developmental biologists and epigeneticists who focus on these processes. As the authors note, cytoskeletal structure and ciliary function play a role in multiple human diseases, and as such may be of interest to translational biologists who focus on these diseases.

I respectfully submit these comments as a neurosurgeon-scientist who studies the chromatin structure and epigenetic landscape of pediatric brain tumors.

We thank the reviewer for their time and both thoughtful and positive comments, which helped us to improve the manuscript.

March 7, 2023

RE: Life Science Alliance Manuscript #LSA-2023-02023-T

Prof. Ralph A.W. Rupp
Ludwig-Maximilians-Universität München
Biomedical Center, Molecular Biology
Grosshaderner Strasse 9
Planegg-Martinsried D-82152
Germany

Dear Dr. Rupp,

Thank you for submitting your revised manuscript entitled "The Histone H4K20 methyltransferase Suv4-20h1/KMT5b is required for multiciliated cell differentiation in *Xenopus*". We would be happy to publish your paper in Life Science Alliance pending final revisions necessary to meet our formatting guidelines.

- please add ORCID ID for corresponding author-you should have received instructions on how to do so
- please add a summary blurb/alternate abstract and a category for your manuscript to our system
- please consult our manuscript preparation guidelines <https://www.life-science-alliance.org/manuscript-prep> and make sure your manuscript sections are in the correct order
- please add the Twitter handle of your host institute/organization as well as your own or/and one of the authors in our system
- please make sure that the spelling of all the author names in the manuscript match with the names that are entered in our system
- please add the author contributions to the main manuscript text
- please add a figure callout for Figure 1 F,G and Figure S1 A,B
- please add a Data Availability Statement at the end of the Materials and Methods section to repeat the accession information for the RNA-seq data

Figure Check:

- please make sure these figures have scale bars with sizes indicated: Figure 1B; Figure 3A; Figure 5A,C; Figure 6A; Figure S1B,D; Figure S2A,D; Figure S4A,D

A. FINAL FILES:

B. MANUSCRIPT ORGANIZATION AND FORMATTING:

Sincerely,

April 18, 2023

RE: Life Science Alliance Manuscript #LSA-2023-02023-TR

Prof. Ralph A.W. Rupp
Ludwig-Maximilians-Universität München
Biomedical Center, Molecular Biology
Grosshaderner Strasse 9
Planegg-Martinsried D-82152
Germany

Dear Dr. Rupp,

Thank you for submitting your Research Article entitled "Histone methyltransferase SUV4-20H1/KMT5b is required for *Xenopus* multiciliated cell differentiation". It is a pleasure to let you know that your manuscript is now accepted for publication in Life Science Alliance. Congratulations on this interesting work.

DISTRIBUTION OF MATERIALS:

Again, congratulations on a very nice paper. I hope you found the review process to be constructive and are pleased with how the manuscript was handled editorially. We look forward to future exciting submissions from your lab.

Sincerely,
